# Magnetically Tuned Impedance and Capacitance for FeGa/PZT Bilayer Composite under Bending and Longitudinal Vibration Modes

**DOI:** 10.3390/s22239283

**Published:** 2022-11-29

**Authors:** Lei Chen, Yao Wang, Fujian Qin, Zhongjie Wan

**Affiliations:** 1Key Lab. of Computer Vision and Intelligent Information System, Chongqing University of Arts and Sciences, Chongqing 402160, China; 2School of Electronic Information and Electrical Engineering, Shanghai Jiao Tong University, Shanghai 200240, China

**Keywords:** magnetostrictive material, magnetoimpedance effect, magnetostrictive strain, magnetic permeability, Young’s modulus, magnetoelectric composite

## Abstract

A magnetically tunable magnetoelectric transducer consisting of rectangular Fe_82_Ga_18_(FeGa)/Pb(Zr,Ti)O_3_(PZT) composites is developed, and their magnetoimpedance and magnetocapacitance effects are investigated under bending and longitudinal modes. Specifically, the composites’ impedance and capacitance are found to vary with dc magnetic field H_dc_, which results from the varied effective dielectric permittivity of the FeGa/PZT composite with H_dc_ due to the delta E effect, magnetostrictive effect of FeGa and mechanism responsible for ME coupling between the FeGa and PZT layers. Furthermore, the FeGa/PZT bilayered composite exhibits both bending and longitudinal vibration modes due to the asymmetrical stress distributions. The maximum ΔZ/Z of the FeGa/PZT composite is about 215% at the antiresonance frequency *f*_a_ = 28.78 kHz of the bending-mode, which is 2.53 times as high as that at the antiresonance frequency *f*_a_ = 107.9 kHz of the longitudinal mode, while the maximum ΔC/C of the FeGa/PZT composite is about 406% at the resonance frequency *f*_r_ = 28.5 kHz of the bending mode, which is 3.5 times as high as that at the antiresonance frequency *f*_a_ = 106.6 kHz of the longitudinal mode. This study plays a guiding role for the design and corresponding application of magnetic sensors, magnetic-field-tuned electronic devices and multiple frequency ultrasonic transducers.

## 1. Introduction

The magnetoelectric (ME) effect, defined as the magnetization induced by an electric field (E) or the polarization induced by a magnetic field (H), is a product property of magnetostrictive and piezoelectric phases of magnetoelectric composites [1,2,3,4]. Correspondingly, ME composites enable effective energy transfer between electric and magnetic energies and exhibit strong ME coupling effects, which has increasingly attracted interest in the last two decades and led to many novel multifunctional device applications such as in magnetic field sensors, memory devices, power generators and transformers [5,6,7,8,9,10].

Recently, ME composites have also been used in electronic switching device applications due to the tunable capacitance or impedance with applied dc bias magnetic field (H_dc_), which is termed as the magnetocapacitance (MC) or magnetoimpedance (MI) effect [11,12,13,14,15]. When a dc bias magnetic field H_dc_ is applied to an ME composite, the magnetostrictive phase produces an elastic deformation due to the piezomagnetic effect, and then the induced strain alters the dielectric permittivity and capacitance of the piezoelectric phase piezoelectric effect due to the stress–strain coupling of the interlayer. It is noted that the MI effect of ME composites differs from the conventional giant MI effect of soft magnetic materials, since the former is determined by both the permeability and permittivity of the ME composite, while the latter is mainly affected by the skin effect and magneto-inductance of soft magnetic material [16]. Additionally, the conventional dc magnetic field sensor based on the ME effect usually needs an extra superimposed alternating current magnetic field to resonate the device in addition to the sensed dc magnetic field. Accordingly, it takes additional power consumption and excitation equipment to generate the ac magnetic field. For comparison, the MI and MC effects of ME composites can be utilized to sense the dc magnetic field without superimposing an additional alternating current magnetic field, which reduces the power consumption of ME devices and facilitates their miniaturization.

Recently, many researchers have studied the MC and MI effects of single-phase materials and ME composites intensively. For example, Wan et al. observed a room-temperature MC effect as large as 45% at H_dc_ = 6.0 kOe and an electrical excitation frequency of 1 kHz in the single-phase film of Sn_0.98_Co_0.02_O_2_ [14]. Singh et al. presented 0.70BNT–0.30NFO composites synthesized by the sol–gel method that exhibited a 27.51% MC effect at 1 kHz frequency [13]. Castel et al. proposed a 10% MC effect in BaTiO3/Ni nanocomposites at a high H_dc_ of 2 kOe [17,18]. Ramachandran reported the magnetoimpedance (130%) effect of single-phase 45PMN-20PFW-35PT at an H_dc_ of 5 kOe [19]. Fina et al. presented a nanocomposite of BaTiO_3_/CoFe_2_O_4_ that exhibited a capacitance change up to 2% at H_dc_ = 8 T [12]. However, most of these studies only exhibit a weak MC or MI effect at room temperature, which is not suitable for practical applications. In order to improve the magnetically tunable MC effect, Wang et al. investigated the giant magnetocapacitance effect of a Terfenol-D/PZT/Terfenol-D composite at a high applied magnetic field of 1.5 kOe [20]. Zhang et al. studied a nonlinear model for the magnetocapacitance effect of PZT-ring/Terfenol-D-strip composites and discussed the effect of boundary conditions on magnetocapacitance [21,22]. However, Terfenol-D possesses a quite low relative magnetic permeability (μ_r_ = 10) and mechanical brittleness. Moreover, it requires a very large magnetic field to tune the magnetoimpedance, magnetocapacitance and magnetoinductance effects, which increases the difficulty of tuning. One possible approach to improve these parameters is to use an Fe_82_Ga_18_ alloy (i.e., Galfenol) with a large magnetic permeability (μ_r_ = 180), excellent mechanical ductility, high blocking stress and low cost. Based on the above consideration, the FeGa alloy was chosen as the magnetostrictive material.

ME coupling can be greatly enhanced by operating the ME composite at the resonance frequency. Therefore, giant MC or MI effects can be also achieved at the resonance conditions. However, previous studies [14,20] mainly focused on the varied capacitance, inductance and impedance of symmetric trilayered ME composites by applying the dc magnetic field at the longitudinal resonance frequency. In fact, compared with the symmetric trilayered ME composite, the asymmetric bilayered ME composite can not only operate in the longitudinal vibration mode, but also in the bending vibration mode due to the asymmetric stress distribution in the ME composite. Correspondingly a bilayered composite operating in a bending mode can work at a lower resonance frequency, or with reduced sample dimensions at the same frequency, compared to a trilayered ME composite. Furthermore, compared to the trilayered ME composite, the asymmetrical bilayered ME composite is able to generate much larger magnetoimpedance, magnetocapacitance and magnetoinductance effects at the low-frequency bending resonance frequency. However, the corresponding physical effect and analysis have rarely been studied.

In this study, an asymmetric bilayered ME composite consisting of piezoelectric ceramic PZT and FeGa alloy is prepared. The magnetoimpedance and magnetocapacitance effects of the FeGa/PZT bilayered composite are investigated. The FeGa/PZT bilayered composite works under both bending and longitudinal vibration modes due to its asymmetric configuration. Correspondingly, the ΔZ/Z and ΔC/C of the FeGa/PZT composite are analyzed and compared under different modes. Furthermore, the dependences of resonance frequency, impedance and capacitance on H_dc_ are discussed. This study facilitates the understanding of ME composites’ electrical behavior and plays a guiding role at the design of magnetic sensors, magnetic field-tuned electronic devices and multiple frequency ultrasonic transducers.

## 2. Experiment

Figure 1 demonstrates the schematic illustration of the asymmetric bilayered FeGa/PZT composite. The ME composite is a plate type FeGa/PZT laminated composite, which operates in the L–T (i.e., longitudinal–transverse) mode. The magnetostrictive material FeGa was longitudinally magnetized along the length direction, and the piezoelectric material PZT was transversely poled along the thickness direction. The magnetostrictive FeGa alloy (Fe_82_Ga_18_ or Galfenol) was produced by Suzhou A-one Special Alloy, Ltd., China, and its size was 12 × 6 × 0.8 mm^3^. FeGa alloy exhibits good ductility, high mechanical strength and large magnetostriction at the low saturation field. The PZT (Pb(Zr,Ti)O_3_) plate with dimensions of 12 mm × 6 mm × 0.8 mm was commercially supplied by Electronics Technology Group Corporation No. 26 Research Institute, Chongqing, China. The upper and lower surfaces of the PZT plate were coated with Ag electrodes. The asymmetric bilayered FeGa/PZT composite was fabricated by stacking and bonding the FeGa plate on the bottom side of the rectangular PZT plate with commercial epoxy glue (E-Solder No. 3021, ACME Division of Allied Products Co., Schenectady, NY, USA). Then, the ME composites were pressed using a hydraulic press and cured at 80 °C for 4 h to minimize the adhesive layer thickness and guarantee strong bonding.

When analyzing the magnetoimpedance and magnetocapacitance effects of the ME composite, an Impedance Analyzer (4194A HP Agilent) setup was used to measure the FeGa/PZT composite’s impedance Z and capacitance C at the resonance frequencies of the bending and longitudinal vibration modes. In the experiment, a dc bias magnetic field (H_dc_) was generated by a pair of electromagnets driven by an SR830 Lock-In Amplifier. A pair of electromagnets were located at both sides of the FeGa/PZT composite along the length direction of the samples. The magnitude of H_dc_ was calibrated by a Gauss magnetometer (Lake Shore 455 DSP) in real time, ranging from 0 Oe to 1500 Oe. The impedance Z and capacitance C of FeGa/PZT composite were measured with a precision impedance analyzer. The acquired data were stored in a computer with the LabVIEW VI program. The corresponding measurement setup is illustrated in Figure 2. Specifically, the impedance spectrum was measured at different dc magnetic fields by using a frequency-swept method around the resonance frequency. All the experiments were carried out at room temperature and ambient pressure.

## 3. Results and Discussion

### 3.1. Magnetoimpedance Effect

As the structure of the FeGa/PZT bilayered composite is asymmetric, the asymmetrical stress distributions of the magnetostrictive and piezoelectric layers produce flexural deformation and result in the bending of the FeGa and PZT layers. Correspondingly, the bending vibration mode can be also observed for the FeGa/PZT composite in addition to the longitudinal vibration mode existing in the symmetrical trilayered composites. Here, the bending resonance frequency of the FeGa/PZT bilayered composite can be expressed by [23].
(1)$$f_{b}=\frac{\pi d}{4\sqrt{3}l^{2}}\sqrt{\frac{\bar{E}}{\bar{\rho }\;}}\beta_{n}^{2}$$
where $\beta_{n}=n+1/2$ and *n* is the order of the bending vibration mode. When *n* = 1, it is the first-order bending vibration mode. *d* and *l* are the thickness and length of the composite, respectively. The average density $\bar{\rho }$ and average Young’s modulus $\bar{E}$ of the FeGa/PZT laminated composite can be expressed as
(2)$$\bar{\rho }=m\rho_{p}+\left(1-m\right)\rho_{f}$$
(3)$$\bar{E}=mE_{p}+\left(1-m\right)E_{f}$$
where *m* and 1 − *m* are the volume fractions of the PZT and FeGa layers, respectively; $E_{p}$ and $E_{f}$ are the Young’s moduli of the PZT and FeGa layers, respectively; and $\rho_{p}$ and $\rho_{f}$ are the densities of the PZT and FeGa layers, respectively. The longitudinal resonance frequency is evaluated by [23]
(4)$$f_{L}=\frac{1}{2l}\sqrt{\frac{\bar{E}}{\bar{\rho }\;}}$$

By using the typical material parameters of the bilayered composite listed in Table 1, Equations (1) and (4), and the geometries of the laminate given above, the first bending resonance frequency (*n* = 1) and longitudinal resonance frequency of the FeGa/PZT bilayered composite are predicted to be 28.9 kHz and 106.5 kHz, respectively. Here, the first bending resonance frequency is much lower than the first longitudinal resonance frequency.

The impedance and phase angle spectra of the FeGa/PZT composite are measured over a frequency range of 1 kHz < f < 120 kHz, as is shown in Figure 3. Two minimum impedances of 9.5 and 2.3 kΩ can be observed at the electromechanical resonances of 28 kHz and 106 kHz, respectively. Here, the resonance peak at 28 kHz is ascribed to the first bending mode, while the resonance peak at 106 kHz is attributed to the first longitudinal resonance mode. Such experimental results agree with the predicted results well.

Next, the magnetoimpedance effects of the FeGa/PZT composites under both bending and longitudinal vibration modes were measured as a function of electrical excitation frequency at various dc magnetic fields (H_dc_), as shown in Figure 4 and Figure 5. Firstly, the FeGa/PZT composite operates in the bending vibration mode due to the flexural deformation caused by asymmetric magnetostrictive stress in the bilayered composite. It can be seen in Figure 4 that when H_dc_ = 3 Oe, the impedance reaches the minimum value (i.e., Z_n_ = 2.2 kΩ) at the bending resonance frequency *f*_r_ = 28.1 kHz and the maximum value (i.e., Z_m_ = 9.1 kΩ) at the bending antiresonance frequency *f*_a_ = 28.5 kHz. For comparison, the minimum impedance (i.e., Z_n_ = 0.55 kΩ) and maximum impedance (i.e., Z_m_ = 2.35 kΩ) of the FeGa/PZT composite under the same magnetic field of 3 Oe occur at the longitudinal resonance frequency *f*_r_ = 106.7 kHz and antiresonance frequency *f*_a_ = 107.9 kHz, respectively. It should be noted that the resonance and antiresonance frequencies of the FeGa/PZT composites under the bending and longitudinal vibration modes have great discrepancies due to the different vibration behaviors.

Furthermore, it is interesting to find that resonance frequency f_r_ and antiresonance frequency f_a_ of FeGa/PZT composites show obvious dependences on H_dc_, as shown in Figure 6. Specifically, f_r_ and f_a_ vary with the dc magnetic field in a similar trend for the ME composite operating at the same vibration mode; however, the trends of f_r_ and f_a_ as a function of H_dc_ are distinct between the bending mode and longitudinal mode. For the FeGa/PZT composite at the bending mode, f_r_ and f_a_ decrease to a minimum value with the increasing H_dc_, and then gradually increase when H_dc_ further increases. For comparison, the f_r_ and f_a_ of the FeGa/PZT composite at the longitudinal mode increase with increased H_dc_ until reaching the maximum saturation value at H_dc_ = 800 Oe. The obvious nonmonotonic trend of the resonance frequency with H_dc_ is attributed to the varied Young’s modulus caused by magnetostriction (i.e., the ΔE effect). Specifically, the Young’s modulus *E*_m_ of FeGa varies with H_dc_ due to different magnetic domain movements under various magnetic fields. According to Equations (1) and (4), the resonance frequency of the ME composite is proportional to the ME composite’s average Young’s modulus, which results in the dependence of H_dc_ on the resonance frequency. This study is expected to guide the design of magnetic field-tuned ultrasonic transducers.

It is also found that the maximum impedance (Z_m_) corresponding to the antiresonance frequency f_a_ and the minimum impedance (Z_n_) corresponding to the resonance frequency f_r_ vary nonmonotonically with H_dc_, as shown in Figure 2 and Figure 3. When H_dc_ increases from 0 Oe to 1500 Oe, the maximum impedance Z_m_ of the bending-mode FeGa/PZT composite varies from 7.02 kΩ to 10.22 kΩ and the maximum impedance Z_m_ of the longitudinal-mode FeGa/PZT composite changes from 21.01 kΩ to 25.62 kΩ. This means that the impedance of the FeGa/PZT laminated composite can be magnetically tuned. The reason for this is that the impedance is inversely proportional to the relative effective permittivity. According to the constitutive equations of the piezoelectric and magnetostrictive phases and the interlayer elastic coupling [24], the relative effective permittivity $\varepsilon_{\mathrm{eff}}$ can be given by
(5)$$\varepsilon_{\mathrm{eff}}=\varepsilon_{33,p}+\frac{d_{31,p}^{2}}{s_{11,p}}\left[\frac{n\left[s_{11,m}-\left(q_{11,m}^{2}/\mu_{11}\right)\right]}{\left(1-n\right)s_{11,p}+n\left[s_{11,m}-\left(q_{11,m}^{2}/\mu_{11}\right)\right]}\times \frac{2\tan\left(\frac{kl}{2}\right)}{\frac{kl}{2}}-1\right]$$
where $k=\omega \sqrt{\bar{\rho }{\left[(n/s_{11,p})+\left(\left(1-n\right)/\left(s_{11,m}-\left(q_{11,m}^{2}/\mu_{11}\right)\right)\right)\right]}^{-1}}$; ω=2πf is the angular frequency; *kl*/2 = π/2 is the resonant condition; $d_{31,p}^{}$ and $\varepsilon_{\mathrm{eff}}$ are the piezoelectric coefficient and relative effective permittivity of the piezoelectric material, respectively; $q_{11,m}$ is the piezomagnetic coefficient of the magnetostrictive material; $s_{11,m}$ and $s_{11,p}$ are the compliance coefficient of the magnetostrictive and piezoelectric materials, respectively; $\bar{\rho }$ is the average mass density; and *n* is the piezoelectric material volume fraction. Equation (5) shows that the $\varepsilon_{\mathrm{eff}}$ of the ME composite is closely relative to the piezomagnetic coefficient $q_{11,m}$, compliance coefficient $s_{11,m}$ and the angular frequency. For the FeGa/PZT laminated composite, the applied dc magnetic field induces the magnetostriction of FeGa due to the varying piezomagnetic effect with the dc magnetic field H_dc_. Such strains are transferred to the neighboring piezoelectric material PZT due to the interface coupling, which changes the dielectric polarization of the FeGa/PZT composite. Furthermore, the applied dc magnetic field produces the varied Young’s modulus of FeGa due to the ΔE effect, which results in the variations of the resonance frequency and compliance coefficient $s_{11,m}$. Such magnetic field-induced changes of the piezomagnetic coefficient, the resonance frequency and the compliance coefficient cause the varied $\varepsilon_{\mathrm{eff}}$ of the FeGa/PZT composite with H_dc_, which in turn leads to a change of impedance considering that the impedance of the ME composite is inversely proportional to the relative effective permittivity $\varepsilon_{\mathrm{eff}}$.

To further investigate the magnetoimpedance (MI) ratio with H_dc_, the MI ratios of the FeGa/PZT composite were measured at f_a_ and f_r_ of both the resonant and antiresonant frequencies (Figure 7), respectively. Here, the MI ratio is defined as $\frac{\Delta Z}{Z}=\frac{\left|Z\left(H_{\mathrm{dc}}\right)-Z\left(H_{\max}\right)\right|}{Z\left(H_{\max}\right)},$ where $Z\left(H_{\mathrm{dc}}\right)$ and $Z\left(H_{\max}\right)$ are the impedance values at H_dc_ and the maximum field $H_{\max}$ of 1500 Oe, respectively. It was found that the maximum MI ratio ΔZ/Z of the bending-mode FeGa/PZT composite at the antiresonance frequency f_a_ = 28.78 kHz and resonance frequency f_r_ = 28.5 kHz is about 215% and 115.8%, respectively. For comparison, the maximum ΔZ/Z of the longitudinal-mode FeGa/PZT composite is about 85% and 11.67% at the antiresonance frequency f_a_ = 107.9 kHz and resonance frequency f_r_ = 106.6 kHz, respectively. Compared with the longitudinal-mode FeGa/PZT composite, the ΔZ/Z of the bending-mode FeGa/PZT composite is much higher at both the resonance and antiresonance frequencies, respectively. Specifically, the maximum ΔZ/Z of the bending-mode FeGa/PZT composite is 2.53 times as high as that of the longitudinal-mode FeGa/PZT at the antiresonance frequency. The reason is as follows: on the one hand, the longitudinal resonance frequency is about 4 times higher than the bending resonance frequency, and the compliance coefficient is inversely proportional to the resonance frequency. Hence, the compliance coefficient of the FeGa/PZT composite at the bending mode is much higher than that at the longitudinal mode. It is known that the ME composite’s impedance is determined by the compliance coefficient, since the obvious change of the compliance coefficient at the bending mode with the dc magnetic field causes the significant variations of the relative effective permittivity $\varepsilon_{\mathrm{eff}}$ and impedance with increased H_dc_. On the other hand, the applied $H_{\mathrm{dc}}$ along the length direction of FeGa induces the magnetostriction, and the magnetostrictive strain is transferred to the neighboring piezoelectric material PZT due to the interface coupling effect. When the FeGa elongates, the PZT shortens at first due to the bending strain caused by the modulus discrepancy of FeGa and PZT and the asymmetric structure of the bilayered FeGa/PZT composite. In this case, the bilayer FeGa/PZT laminated structure will be bent for the bending vibration mode. Due to the nonsymmetrical stress distribution in the bilayered FeGa/PZT composite, the bending strain varies more obviously under various H_dc_ compared to that at the longitudinal mode. Correspondingly, the bilayered ME composite operating at the bending vibration mode exhibits a lower resonance frequency and stronger magnetoimpedance effect compared with that at the longitudinal vibration mode. Hence, the bilayered bending-mode FeGa/PZT composite not only exhibits a giant ΔZ/Z effect, but also keeps its size small and decreased eddy current loss compared to the longitudinal-mode ME composite at the high frequency. This result facilities the optimization of ME composites with a giant magnetoimpedance effect. Furthermore, the magnetically modulated impedances of FeGa/PZT composites provide wide applications for tunable spin filters, memory devices, magnetic anomaly sensing, etc.

### 3.2. Magnetocapacitance Effect

The capacitance with two plain plates is $C=\frac{A}{d}\varepsilon_{\mathrm{eff}}$, where *A* is the area of a plate and *d* is the thickness of the piezoelectric material. For the ME composite, the expression of the capacitance can be written as [24]
(6)$$C=\frac{A}{d}\varepsilon_{33,p}+\frac{A}{d}\frac{d_{31,p}^{2}}{s_{11,p}}\left[\frac{n\left[s_{11,m}-\left(q_{11,m}^{2}/\mu_{11}\right)\right]}{\left(1-n\right)s_{11,p}+n\left[s_{11,m}-\left(q_{11,m}^{2}/\mu_{11}\right)\right]}\times \frac{2\tan\left(\frac{\mathrm{kl}}{2}\right)}{\frac{\mathrm{kl}}{2}}-1\right]\;$$

From Equations (5) and (6), the capacitance *C* depends on the relative effective permittivity *ε_r_*. The magnetic field-induced change of piezomagnetic coefficient $q_{11,m}$ and compliance coefficient $s_{11,m}$ in the magnetostrictive material leads to the variations of the relative effective permittivity *ε_r_*. Correspondingly, this leads to the change of capacitance with the dc magnetic field, namely the magnetocapacitance (MC) effect. Castel et al. have reported BaTiO_3_/Ni nanocomposites which show a giant magnetodielectric effect or magnetocapacitance effect [17,18].

At electromechanical resonance frequency, a dramatic variation of $\varepsilon_{\mathrm{eff}}$ is induced by the enhancement of the stress. Due to the dependence of the MC effect on $\varepsilon_{\mathrm{eff}}$, it is expected to show a giant MC effect at electromechanical resonance. This means that capacitance is a function of applied bias magnetic field intensity, as shown in Figure 8. The capacitance *C* values first decrease and then increase at the bending and longitudinal vibration modes’ resonant frequency with the increase of the magnetic field, then capacitance decreases again with further increased H_dc_. Moreover, capacitance values have a smaller change at the dual-mode antiresonance frequency with the increasing H_dc_. However, the impedance Z and capacitance C exhibit opposite trends as a function of the dc magnetic field. This is because the impedance is inversely proportional to the relative effective permittivity $\varepsilon_{\mathrm{eff}}$ or corresponding capacitance, while it is directly proportional to the relative permeability *μ_r_* and corresponding inductance of the ME composite. At the resonance frequency, the capacitance achieves the maximum value while the inductance reaches the minimum value, which leads to the minimum impedance, and vice versa [25]. This also indicates that the MC effect of the ME composite caused by strain induces magnetic and ferroelectric domain interactions.

The magnetocapacitance ratio (MC) is defined as follows [16]:(7)$$\Delta C/C=\frac{\left|C\left(H_{\mathrm{dc}}\right)-C\left(H_{\max}\right)\right|}{C\left(H_{\max}\right)}$$
where $C\left(H_{\mathrm{dc}}\right)$ and $C\left(H_{\max}\right)$ are the sample capacitance values at the dc bias magnetic fields and the maximum field $H_{\max}$ of 1500 Oe, respectively. To precisely observe the magnetic field-dependent magnetocapacitance at different frequencies, magnetocapacitance versus magnetic fields curves under different frequencies are shown in Figure 9.

At the resonant frequency of 28.5 kHz, the magnetocapacitance ratio ΔC/C of the bending-mode FeGa/PZT composite reaches the maximum value of 406% at the magnetic field of 0.8 kOe. Compared with the previously reported room-temperature MC of 45% at the frequency of 1.0 kHz and saturated magnetic field of 6.0 kOe in Co^2+^ doped SnO_2_ dielectric films [14] as well as the maximum MC of 200% for the piezoelectric–magnetostrictive resonator at the anti-resonance frequency [11], the room-temperature MC achieved by this study is improved significantly. In the same case, the maximum MC of the longitudinal-mode FeGa/PZT composite reaches 115% at the resonant frequency of 106.6 kHz. Such strong abilities of tuning capacitance with the magnetic field make the FeGa/PZT composite an ideal candidate for tunable impedance device applications. Additionally, both in the longitudinal-mode and bending-mode condition, the maximum ΔC/C of the FeGa/PZT composite at resonance is larger than that of the FeGa/PZT composite at antiresonance. As previously mentioned, the change trend of impedance and capacitance with the magnetic field is opposite. At resonance, the FeGa/PZT composite is considered to act as a capacitor, where its capacitance reaches a maximum value and its impedance exhibits a minimum value.

## 4. Conclusions

In this study, the impedance and capacitance curves of an FeGa/PZT bilayered composite as a function of dc bias magnetic field is measured. Due to the flexural deformation caused by the asymmetric stress distribution in the ME composite, the FeGa/PZT bilayered composite can not only operate in the longitudinal vibration mode, but also in the bending vibration mode. Furthermore, it exhibits a strong magnetic effect on the electrical impedance and capacitance due to magnetostriction of the magnetostrictive material FeSiB and the effective permittivity of the FeGa/PZT composite with H_dc_ under bending and longitudinal vibration modes. The maximum ΔZ/Z of the bending-mode FeGa/PZT composite is 2.53 times as high as that of the longitudinal-mode FeGa/PZT at the antiresonance frequency, while the maximum ΔC/C of the bending-mode FeGa/PZT composite is 3.5 times as high as that of the longitudinal-mode FeGa/PZT at the resonance frequency. This study plays a guiding role for ME sensor design and its application.

## Figures and Tables

**Figure 1 sensors-22-09283-f001:**
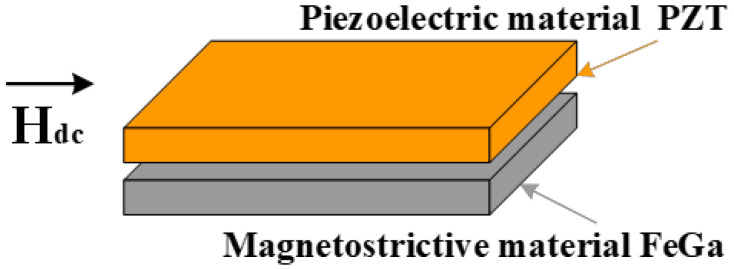
Schematic illustration of the asymmetric bilayered FeGa/PZT composite.

**Figure 2 sensors-22-09283-f002:**
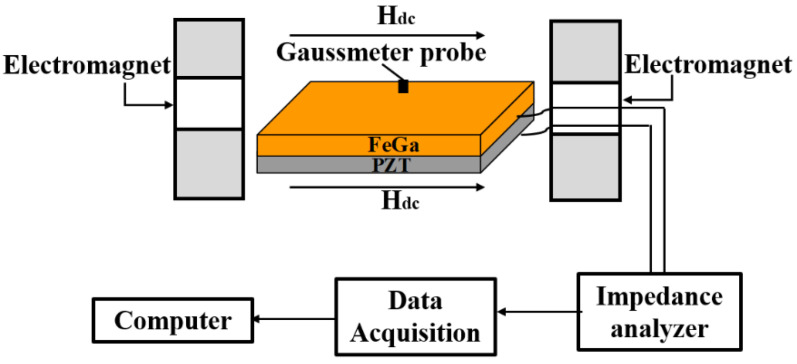
The measurement setup for MI and MC effects in FeGa/PZT composite.

**Figure 3 sensors-22-09283-f003:**
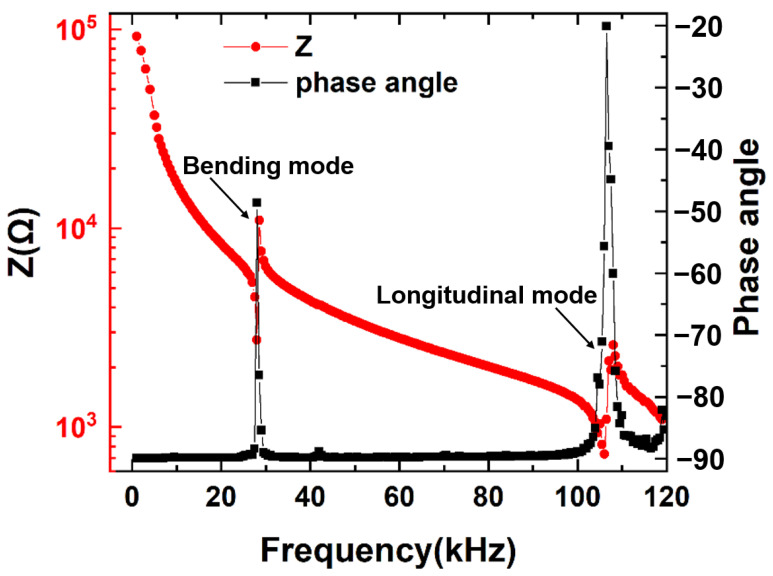
The impedance Z and phase angle of FeGa/PZT laminated composite as a function of frequency under 5 Oe bias magnetic fields.

**Figure 4 sensors-22-09283-f004:**
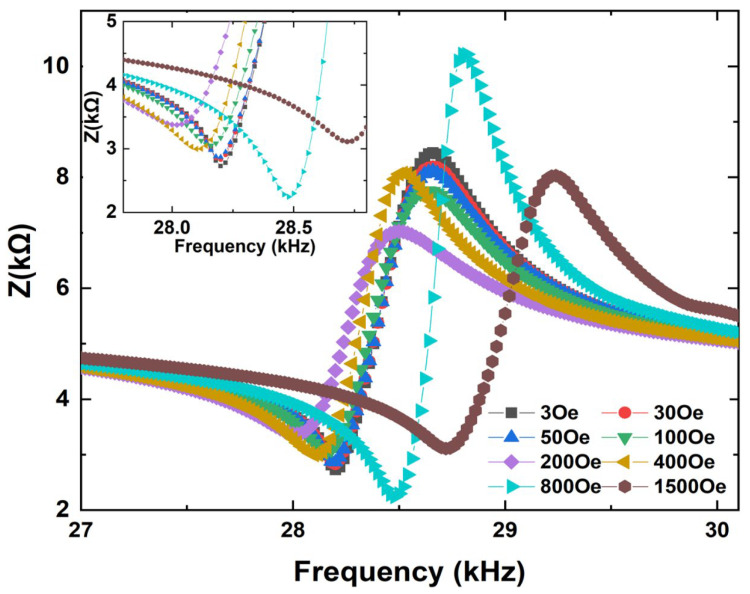
Impedance curve of the bending-mode FeGa/PZT laminated composite at various bias dc magnetic fields in the frequency range of 27 kHz to 30.1 kHz; the insets show enlarged details near the minimum impedances.

**Figure 5 sensors-22-09283-f005:**
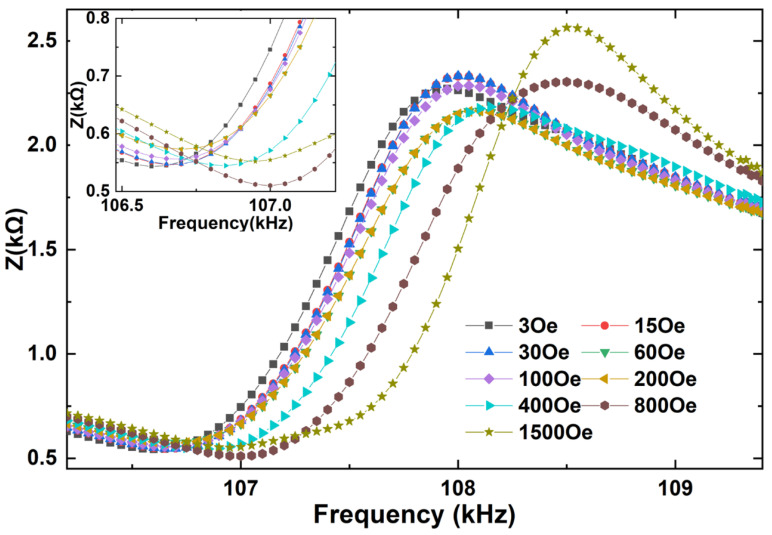
Impedance curve of the longitudinal-mode FeGa/PZT laminated composite at various bias dc magnetic fields in the frequency range between 106.3 kHz and 109.4 kHz; the insets show enlarged details of minimum impedances.

**Figure 6 sensors-22-09283-f006:**
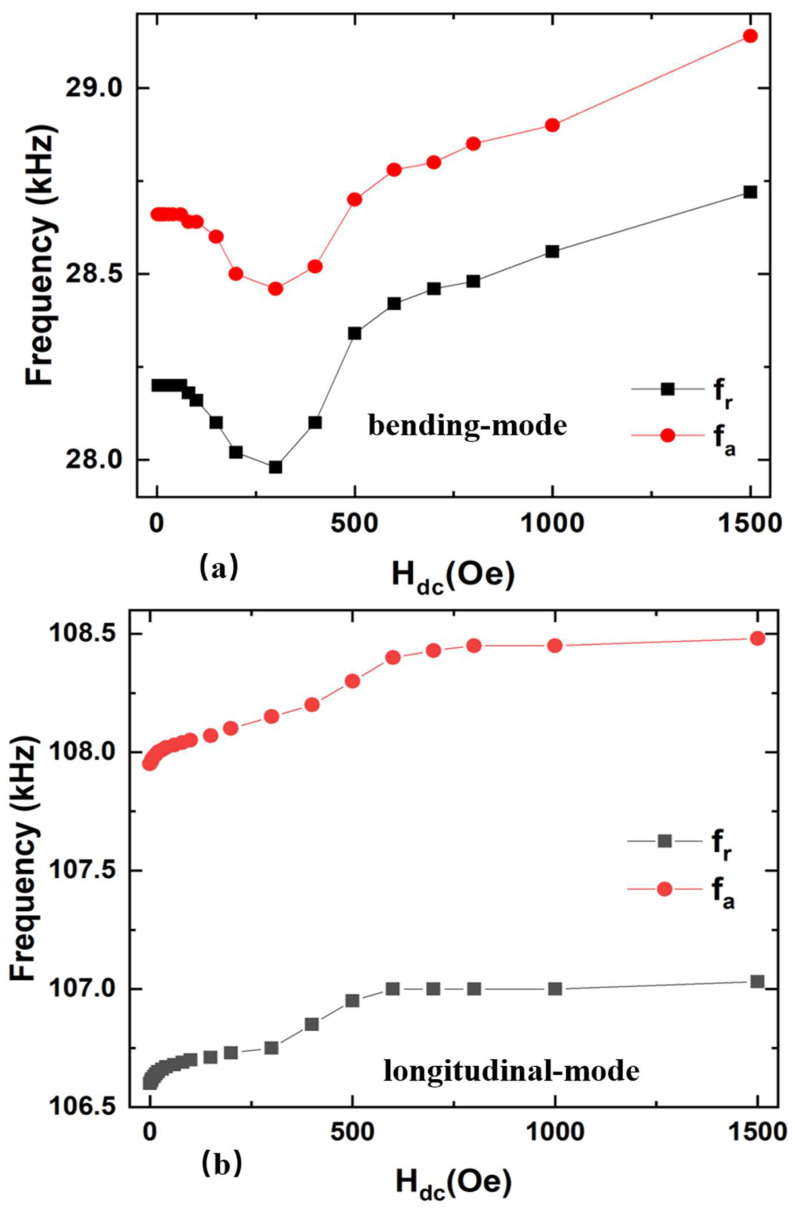
The resonance frequency f_r_ and antiresonance frequency f_a_ as a function of dc magnetic field H_dc_ for FeGa/PZT composite at the (**a**) bending mode and (**b**) longitudinal mode, respectively.

**Figure 7 sensors-22-09283-f007:**
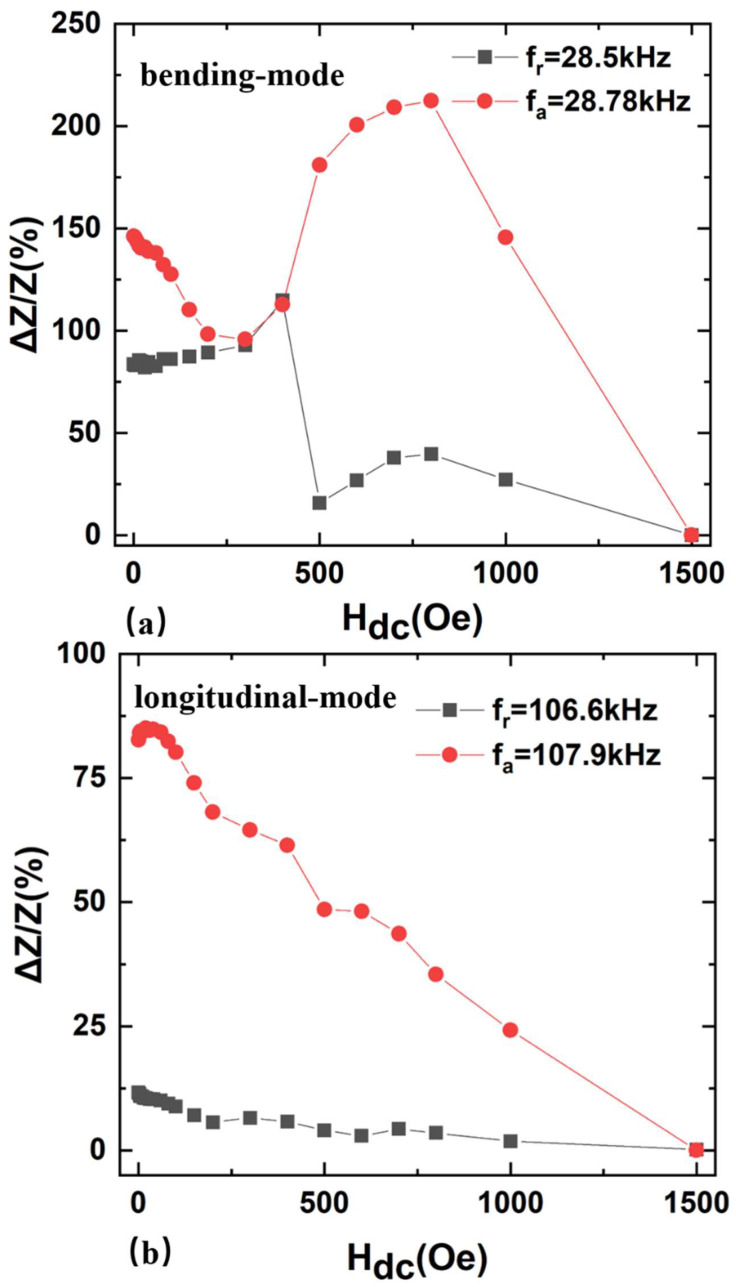
The magnetoimpedance ratios as a function of applied dc magnetic fields for FeGa/PZT composite (**a**) bending mode and (**b**) longitudinal mode.

**Figure 8 sensors-22-09283-f008:**
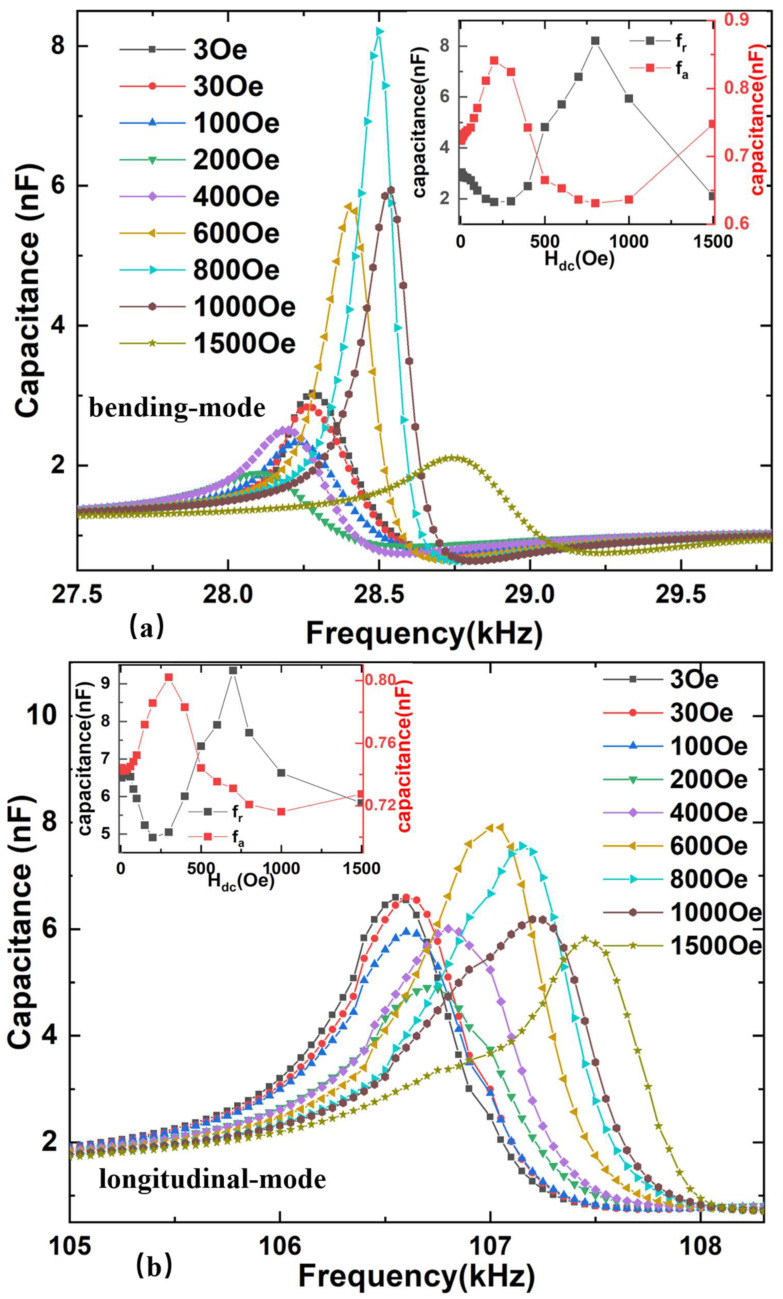
Experimental frequency-dependent capacitances at various dc bias magnetic fields for FeGa/PZT composite (**a**) bending mode and (**b**) longitudinal mode.

**Figure 9 sensors-22-09283-f009:**
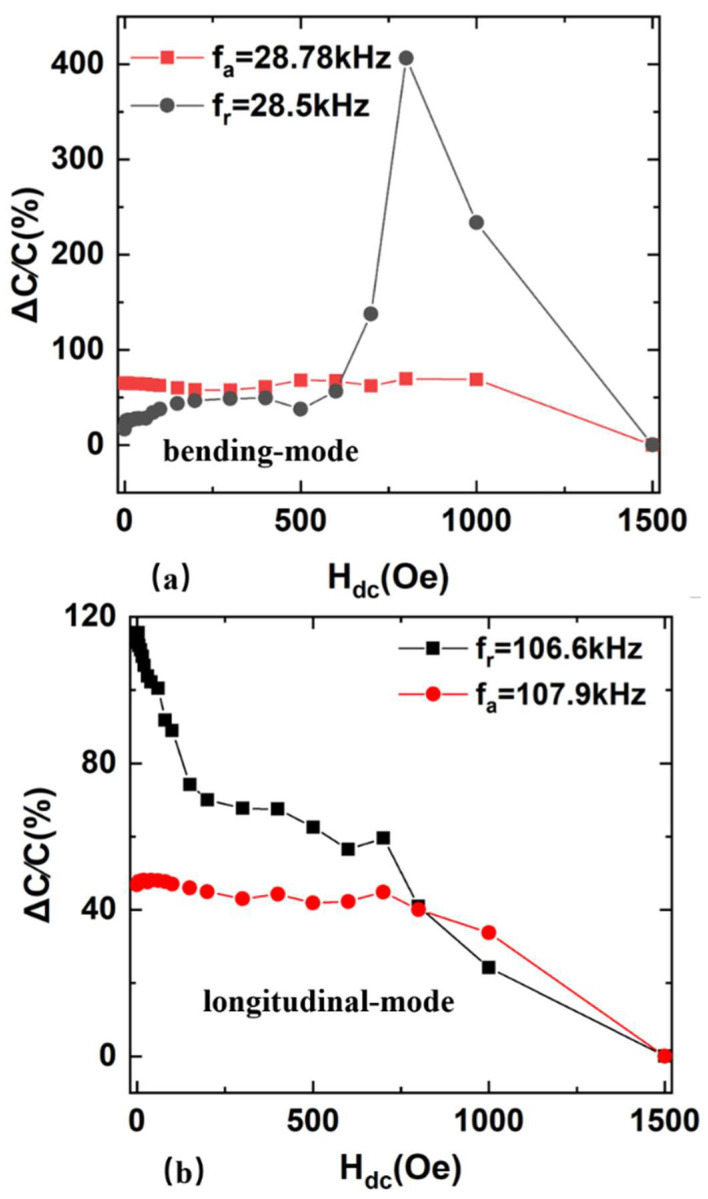
The magnetocapacitance ratios as a function of applied dc magnetic fields for FeGa/PZT composite (**a**) bending mode and (**b**) longitudinal mode.

**Table 1 sensors-22-09283-t001:** The material characteristics of FeGa and PZT.

Material	Density (kg/cm)	Volume Fraction	*μ_r_*	Young’s Modulus (GPa)	*λ_s_* (ppm)
FeGa	7700	50%	180	36	300
PZT	7600	50%		64	

## Data Availability

The data that support the findings of this study are available from the corresponding author upon reasonable request.

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
