# Peer review of "Magnetically Tuned Impedance and Capacitance for FeGa/PZT Bilayer Composite under Bending and Longitudinal Vibration Modes"

_sensors, 2022, doi:10.3390/s22239283_

Round 1

Reviewer 1 Report

Reviewer's report #

            The manuscript entitled “Magnetically tuned impedance and capacitance for FeGa/PZT bilayer composite under bending and longitudinal vibration modes" submitted by Lei Chen, Yao Wang, Fujian Qin, Zhongjie Wan has been reviewed thoroughly. The paper reports the impedance and capacitance properties of bi-layered laminate structures of FeGa alloy with PZT layer. It was mentioned that the physical parameters show variations in their magnitude under the applied magnetic field. In view of this reason, it was also concluded that the ME coupling exists between these two layers, and based on ME effect, it was also proposed in the study that a sensor can be developed and it can be regulated by externally tuning the field. The paper contains original results. Unfortunately, the paper suffers a severe physical interpretation of the obtained results. It just summarizes variations and reports the values obtained. The physics behind the results needs to be discussed thoroughly almost in every result of the paper.

I expect the following specific amendments before it is considered for further processing.

1.       The mechanism responsible for ME coupling between two layers need to be incorporated. For example, please refer to the following papers for information.

Physica status solidi (RRL)–Rapid Research Letters 11 (2017) 1700294; Journal of Physics D: Applied Physics 49 (2016) 405001; Applied Physics Letters 99 (2011) 102506; etc.

2.     The font in inset figures needs to be improved for better visibility.

3.     No specific application related to a particular sensor based on the results needs to be mentioned in the manuscript.

4.     I wonder why the authors vary the magnetic field and measure the electric parameters. Producing a magnetic field itself causes enormous power consumption. The purpose of ME coupling and its usage for low-power consumption applications is defeated. Authors need to clarify the same in the paper, i.e., in the motivation (introduction) part of the manuscript.

In view of the above, I recommend the present paper for major revision.

Reviewer 2 Report

Report on paper "Magnetically tuned impedance and capacitance for FeGa/PZT bilayer composite under bending and longitudinal vibration modes"submitted by Chen et al., for publication in Sensors (sensors-1987517).

The authors proposed a way to tune magnetically the impedance and the capacitance of FeGa/PZT bilayer composite under bending and longitudinal vibration modes. While the paper topic is interesting, the manuscript cannot be accepted in its present form and the authors must perform substantial modifications by addressing the following comments:

  1. In the introduction, the authors should clearly highlight the originality of the paper with respect to the state of the art.
  2. The device should be described in details while including a figure for illustration.
  3. The experimental step-up must be illustrated with a figure while providing its full description including the test conditions.
  4. What are the assumptions used for equations (5) and (6)? Are they justified with respect to the material proprieties and test conditions?
  5. The authors can discuss the applicability of the obtained results in sensing applications. 
  6. What about the sensitivity of the obtained results to temperature variations? 
  7. A comparison in term of capacitance/impedance tunability with respect to other magnetoelectric materials would strengthen the paper highlight.

Round 2

Reviewer 2 Report

The authors have addressed my comments sufficiently to recommend publication of the paper in its current form.